# FADING FOCUS: MITIGATING VISUAL ATTENTION DEGRADATION IN LARGE VISION-LANGUAGE MODELS

## ABSTRACT

How can we ensure that Large Vision-Language Models (LVLMs) maintain strong attention to visual input throughout the inference process? Recent advancements in Large Vision-Language Models (LVLMs) have demonstrated significant progress across multiple domains. However, these models still face the inherent challenge of integrating vision and language for collaborative inference, which often leads to "hallucinations," outputs that are not grounded in the corresponding images. Many efforts have been made to address these challenges, but each approach comes with its own limitations, such as high computational costs or expensive dataset annotation. Worse still, many of them fail to recognize the crucial role of visual attention in guiding the model's response generation. In our research, we identify a key limitation in current LVLMs: the model's diminishing attention to visual input as the number of generated tokens increases, which results in performance degradation. To address this challenge, we propose **I**mage attention-guided **K**ey-value merging c**O**llaborative **D**ecoding (IKOD), a collaborative decoding strategy that generates image-focused sequences using key-value merging. This method derives logits from shorter sequences with higher image attention through key-value merging and combines them with those from the original decoding process, effectively mitigating attention decay. Importantly, IKOD requires no additional training or external tools, making it highly scalable and applicable to various models.

## 1 INTRODUCTION

Recent advancements in Large Language Models (LLMs), such as GPT, LLaMA, and Vicuna (Brown et al., 2020; Touvron et al., 2023; Chiang et al., 2023) have profoundly impacted the development of Large Vision-Language Models (LVLMs), enabling significant progress accross various domains like literature (Yang et al., 2024), agriculture (Zhu et al., 2024a), visual content generation (Zhu et al., 2024b) and robotics (Ding et al.). However, LVLMs face inherent limitations in precisely aligning vision and language modalities for collaborative inference. These shortcomings can lead to LVLMs' trustworthy problems like "hallucinations," where the model generates information not grounded in the images. These problems have led to significant challenges in critical fields such as finance (Kang & Liu, 2023) and medical diagnosis (Chen et al., 2024a), adversely impacting the accuracy and safety of decision-making processes within these systems. Therefore, addressing this issue is crucial for enhancing the performance and reliability of LVLMs. Motivated by the concerns of misalignment between vision and language, various approaches have been proposed to address the issue of misalignment, including instruction tuning (Liu et al., 2023a; Zhao et al., 2023; Lin et al., 2023), post-hoc techniques (Zhou et al., 2023; Yin et al., 2023) and contrastive decoding (Leng et al., 2023; Wang et al., 2024; Zhang et al., 2024). While these methods have demonstrated some success, they often rely heavily on additional datasets, external tools, or computational resources. For instance, post-hoc methods depend on external tools such as pre-trained vision-language models (Liu et al., 2023b) and closed-source large models (Brown et al., 2020), which limits their potential for widespread application and incurs high inference costs. Moreover, many of them are inspired by methods designed specifically for single-modal language models, failing to recognize the crucial role of visual attention in guiding the model's response generation.

To address these challenges, we analyze the relationship between LVLM's performance and its visual attention. Our observations show key limitations in current LVLMs: as the number of generated tokens increases, the model's attention to the image gradually diminishes. Further experiments reveal that this reduction in attention negatively impacts the model's performance. Based on these findings, we propose an **I**mage attention-guided **K**ey-value merging c**O**llaborative **D**ecoding strategy (IKOD), a collaborative decoding strategy that generates image-focused sequences while retaining most of the essential information in the response. This approach involves obtaining logits with high image attention from short sequences through compressing KV Cache and merging them with the logits derived from the original decoding process, which can alleviate the decline in attention. Another advantage of our method is that it requires no additional training and does not rely on external tools.

Our primary contributions can be summarized as follows: (1) We investigate the relationship between Large Vision-Language Models (LVLMs) performance and their visual attention, revealing that as the sequence length increases, the model's attention to the image diminishes. This diminishing attention leads to performance degradation and errors in the generated responses. (2) We introduce IKOD, an image attention-guided key-value merging collaborative decoding strategy. This method endows text sequence with high attention on image using key-value merging and integrates the augmented decoding process with the original decoding process to obtain a more accurate output distribution. (3) IKOD does not require additional training or external tools, which is more easily applicable to various models.

## 2 PRELIMINARIES

In this section, we discuss two fundamental components in Large Vision-Language Models (LVLMs): the inference process and the self-attention mechanism in transformer-based architectures. These concepts are crucial for understanding how LVLMs combine visual and textual information to generate meaningful responses.

### 2.1 INFERENCE IN LVLMs

Large Vision-Language Models (LVLMs) commonly have three key components (Liu et al., 2024c; Dai et al., 2023; Zhu et al., 2023): a vision encoder, a connector and a language model. For the visual input $v$, a pre-trained vision encoder is employed to extract visual features $z_v$. The connector primarily involves two types: the Q-former and the MLP. The Q-former functions as a query-based mechanism that interacts with the visual features and the instruction, generating a set of latent embeddings that capture the task-relevant image features. In contrast, the MLP connector applies a series of fully connected layers to transform the visual features into a representation that can be directly fed into the language model. The aligned visual features can be formulated as follows:

$$x_v = H(x_I, z_v), \tag{1}$$

where $H(\cdot)$ denotes the connector module and $x_I$ is the input instruction. In the inference process, the generated token can be defined as sampled from a probability distribution:

$$p(Y|x, x_v) = \prod_{t=1}^{L} p(y_t|y_{<t}, x, x_v), \tag{2}$$

where $y_{<t}$ represents the sequence of generated tokens up to time step $t-1$, and $x$ is the input text tokens and $L$ is the length of the generated sequence.

### 2.2 SELF-ATTENTION IN TRANSFORMER

Transformers have revolutionized the field of deep learning, particularly in natural language processing, due to their self-attention mechanism. The self-attention mechanism enables the model to capture long-range dependencies and interactions between tokens in a sequence by computing attention scores for each pair of tokens.

For an input sequence of tokens $X = \{x_1, x_2, \dots, x_n\}$, each token $x_t$ is first linearly projected into three vectors: a query $q_t$, a key $k_t$, and a value $v_t$ through learned weight matrices $W_Q$, $W_K$, and $W_V$, respectively:

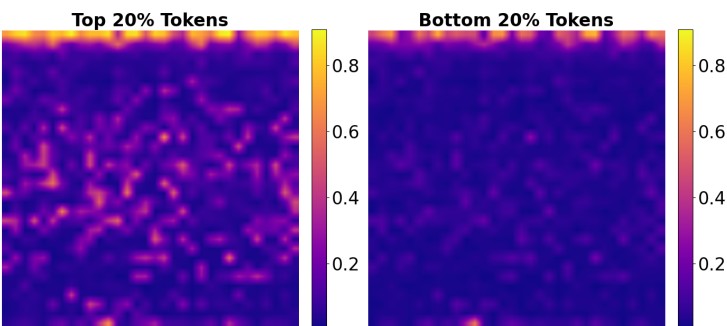

Figure 1: Image attention across different layers and heads of LLaVA 1.5 7b. More examples are avaliable in Appendix  A.2.2

$$Q_t = x_t W_Q, \quad K_t = X_t W_K = [k_1, k_2, \ldots, k_t], \quad V_t = X_t W_V = [v_1, v_2, \ldots, v_t],$$

where $X_t$ represents the entire input sequence when generating $x_t$, both $K_t$ and $V_t$ are concatenations of the keys and values for all tokens in the sequence.

The self-attention mechanism is computed in parallel for all tokens in the sequence by packing the queries, keys, and values into matrices $Q_t$, $K_t$, and $V_t$, respectively. The output of the self-attention mechanism for the entire sequence can be written as:

$$Z_t = \text{softmax}\left(\frac{Q_t K_t^T}{\sqrt{d_k}}\right) V_t, \tag{3}$$

where $Z_t$ represents the matrix of outputs. In addition to the self-attention mechanism, the residual connection (often referred to as a "skip connection") is used to add the input of the previous layer directly to the output of the current layer.

$$Z_{\text{final}} = Z_{\text{prev}} + \text{FFN}(Z_{\text{prev}}), \tag{4}$$

here, $Z_{\text{prev}}$ is the feature from the previous layer, and $\text{FFN}(\cdot)$ is the self-attention network. Consequently, the generated token can be defined as:

$$p(Y|x, x_v) = \prod_{t=1}^{L} p(y_t|y_{<t}, x, x_v) = \prod_{t=1}^{L} p(y_t|Q_t, K_t, V_t). \tag{5}$$

The self-attention mechanism enables transformers to effectively capture long-range dependencies between tokens in a sequence, enhancing the model's ability to understand complex data patterns. However, this architecture for current LVLMs still has a limitation: the model's attention to the image decreases as the token length increases.

## 3 KEY INSIGHT

### 3.1 IMAGE ATTENTION WEAKENS WITH INCREASING SEQUENCE LENGTH

Visual attention within Large Vision-Language Models (LVLMs) has been identified as a distinctive pattern that significantly influences the performance of these models (Lin et al., 2024; Yu et al., 2024a). Inspired by this, we explore the relationship between image attention and token position in the LVLMs' responses.

We randomly sample 5,000 images from the MSCOCO validation dataset (Lin et al., 2014) for our analysis. The prompt used for generating responses is, "Describe this image in detail." Within this context, we examine the correlation between the model's image attention and the token positions in

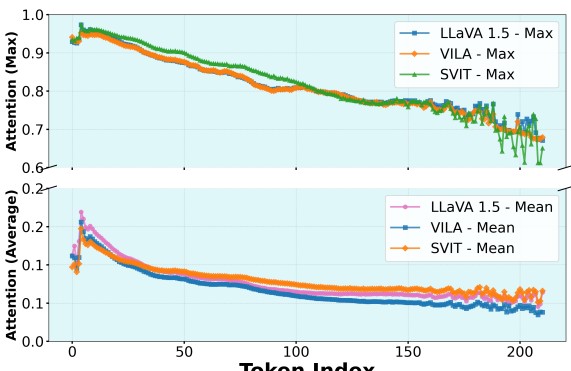

Figure 2: Comparison of average and maximum attention across the generated tokens for LLaVA-1.5, VILA, and SVIT models.

its responses. Specifically, for each token $t$ with the total sequence length $L$, we obtain the attention map at $i$-th layer and $j$-th head by:

$$Att_t^{i,j} = \text{softmax}\left(\frac{Q_t^{i,j}(K_t^{i,j})^T}{\sqrt{d}}\right),\tag{6}$$

where $Q_t^{i,j} \in R^{1\times d}$ and $K_t^{i,j} \in R^{L\times d}$. For better comparison, we select the first 20% and the last 20% of the tokens from each sequence. We then compare the image attention across different layers and heads, analyzing how the attention varies between early and late tokens in the sequence. As shown in Figure 1, we present the visualization of image attention in LLaVA-1.5, where each line represents a different layer. We observe a significant difference between the last 20% of tokens and the first 20%. Specifically, in the last 20%, image attention significantly decreases for most patches. To further validate our findings, we visualize the density distribution of the relationship between attention and token positions using a kernel density estimate (KDE) (The details can be seen in Appendix A.2.1). For each token, we calculate its relative position in the sequence and the average image attention across different heads. As shown in Figure 3, we observe that image attention diminishes as the sequence length increases, which further confirms our findings. We also show average and maximum image attention scores across different heads on different models in Figure 7 in Appendix A.2.2.

## 3.2 Weakened Image Attention Leads to Performance Diminishment in the Model

After observing that the model's image attention weakens as the sequence length increases, we are prompted to consider a question: Does the weakened image attention effect LVLMs' performance? To address this, we conduct a detailed analysis to investigate the relationship between image attention and model performance. The phenomenon of hallucinations in LVLMs refers to instances where these models generate content that is not grounded in the provided image. Such hallucinations are generally viewed as indicators of weak performance in LVLMs, as the generated descriptions or responses deviate from the visual information, leading to inaccuracies in output (Liu et al., 2024a). Therefore, we exam-

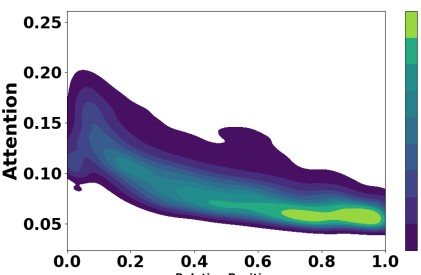

Figure 3: Image attention across different layers and heads of LLaVA-1.5 during response generation, showing the relationship between relative position in the sequence and the average image attention across different heads. More examples can be found in Appendix A.2.2.

ine how visual attention impacts the performance of LVLMs by exploring its connection to the phenomenon of hallucination. Following the setting in 3.1, we visualize the density distribution of the average image attention of tokens and the positions of hallucinated tokens, as illustrated in Figure 4. We conduct experiments on two LVLMs, LLaVA-1.5 and InstructBLIP. As the sequence increases, there's a noticeable pattern where the visual attention decreases, indicating weakened

attention towards tokens appearing later in the sequence. Besides, the hallucinated tokens are more concentrated in areas with low attention, which suggests lower image attention is more likely to cause the model to make errors. Additionally, it's interesting to note that we find InstructBLIP's attention to be much greater than LLaVA's, which may be related to the use of the Q-Former structure. More examples are shown in Appendix A.2.2.

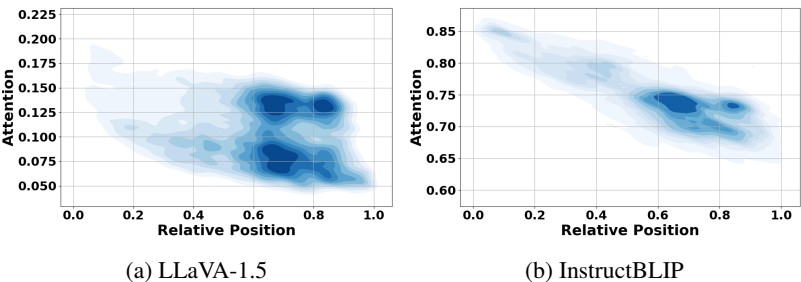

(a) LLaVA-1.5           (b) InstructBLIP

Figure 4: Relationship between image attention and model performance on LLaVA-1.5 and Instruct-BLIP.

## 4   METHOD

While Large Vision-Language Models (LVLMs) have made great progress in integrating visual and textual modalities, they often face challenges in maintaining strong visual attention as sequence length increases. To address these limitations, we propose an Image Attention-Guided key-value merging strategy that selectively integrates key and value vectors based on their importance derived from image attention scores. We first propose the Image Attention-Guided Key-Value Merging Approach in Section 4.1. In Section 4.2, we introduce a collaborative decoding strategy to further enhance the cabilities of LVLMs. Finally, in Section 4.3, we present adaptive plausibility constraints to improve the model's capacity for managing long-sequence image processing. The overall framework of our method is illustrated in Figure 5.

### 4.1   IMAGE ATTENTION-GUIDED KEY-VALUE MERGING

In this section, we propose a key-value merging strategy that prioritizes the integration of visual features by selectively merging key and value vectors based on their importance determined by image attention scores. The core idea is to identify anchor points in the key-value vectors that aggregate surrounding contextual information. By recognizing the significance of visual attention in Large Vision-Language Models (LVLMs), we can develop policies to predict which vectors in the key-value storage will be most relevant for upcoming inference tasks. This approach helps reduce sequence length and mitigates the problem of diminishing image attention.

Our method ensures that LVLM maintains a strong focus on crucial visual elements, thereby improving the quality of generated tokens. During the key-value merging stage, this approach involves two primary steps: 1) selecting important key-value anchors based on the layer-wise sum of image attention scores, and 2) merging vectors based on the selected anchors.

**Anchors Selection.** Suppose the model has $L$ layers in total, each with $K$ heads. The text sequence including instruction and generated tokens has $T$ tokens. Consider the $j$-th attention head in the $i$-th layer, the original key and value are $k^{i,j}$ and $v^{i,j}$ respectively. The attention for the token $y_t$ in text sequence can be denoted as $\text{Att}_t^{i,j}$. Overall, we can calculate the attention score for each token in each layer based on the visual attention, denoted as $S_t^i = \sum_j \text{Att}_t^{i,j}[\text{image\_index}]$, where image_index refers to the index of the image tokens. Consequently, we obtain independent attention scores for each layer. Since we expect all the tokens in text sequence to have higher attention on image, we pay more attention to the tokens with lower attention scores, which commonly appear at the end of sequence and are more relevant with the query token. Thus we select these tokens as anchors to augment them, while merging the remaining tokens' keys and values into the closest anchors'. Notably, we protect the most recent token as it has great association with query token. Then we sort the tokens except the last token (protected token) based on their attention scores reversely for each

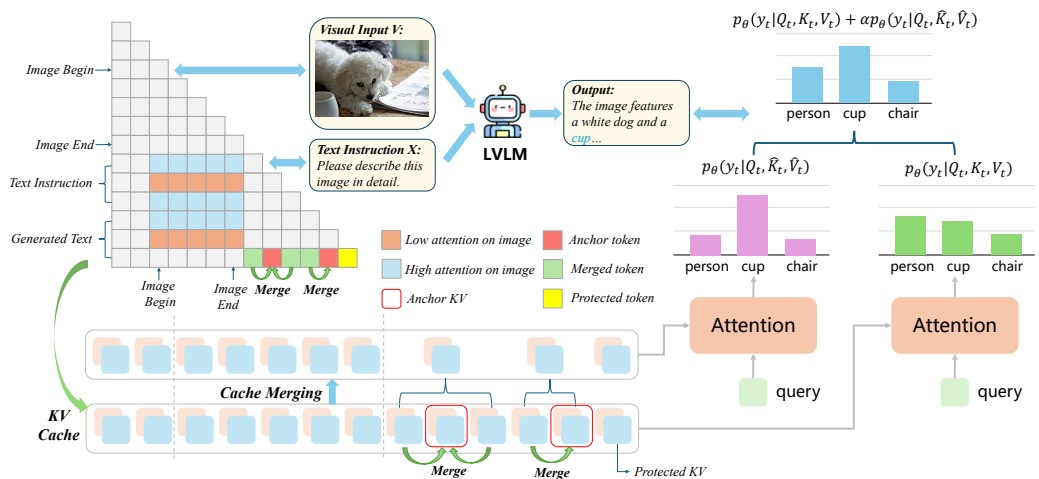

Figure 5: The overall framework of IKOD. We select the tokens with lower attention on image in text sequence to be anchors while merging the remaining tokens' keys and values (KVs) into the closest anchors', resulting in a compressed KV Cache namely a shorter contextual sequence with higher attention on image. Then we combine the logits derived from the compressed KV Cache with the original logits to get a output distribution more grounded in image.

layer, resulting in the indices $\{t_k^i | k = 1, 2, ..., T - 1\}$ in ascending order, where $i$ indicates layer $i$. Given an anchor ratio $\lambda$, the top $K = \lambda \times (T - 1)$ tokens in each layer are selected as anchors, yielding the following set:

$$
D_k^i = \begin{cases} \{0, ..., \left\lfloor \frac{t_1^i + t_2^i}{2} \right\rfloor\}, & k = 1 \\ \{\left\lfloor \frac{t_{k-1}^i + t_k^i}{2} \right\rfloor + 1, ..., \left\lfloor \frac{t_k^i + t_{k+1}^i}{2} \right\rfloor\}, & 1 < k < K , \\ \{\left\lfloor \frac{t_{K-1}^i + t_K^i}{2} \right\rfloor, ..., T - 1\}, & k = K \end{cases} \tag{7}
$$

where $\lfloor \cdot \rfloor$ denotes the floor function. The division indicates that each token is divided into the closest anchor token's group across various layer, attributed to the strong contextual associations of close tokens.

**Key-Value Merging.** When generating the next token, $T + 1$, in each layer, we average all the key-value vectors corresponding to each division $D_k^i$ and merge them into $K_t^{i,j}$ and $V_t^{i,j}$. Specifically, we compute the averaged key and value for the $j$-th head of the $i$-th layer as follows:

$$
\tilde{K}_{t,k}^{i,j} = \frac{1}{|D_k^i|} \sum_{m \in D_k^i} K_m^{i,j}, \quad \tilde{V}_{t,k}^{i,j} = \frac{1}{|D_k^i|} \sum_{m \in D_k^i} V_m^{i,j}, \tag{8}
$$

where $D_k^i$ is the set of all positions in division $k$ for layer $i$, and $|D_k^i|$ represents the number of elements in that division. Next, we concatenate the averaged key and value vectors across all divisions, along with the previous tokens and protected token, to obtain the final merged key and value for the $j$-th head of the $i$-th layer: $\hat{K}_t^{i,j}$ and $\hat{V}_t^{i,j}$.

This approach allows us to obtain a shorter, more image-focused decoding strategy by merging keys and values based on image attention, which can be formulated as $p(y_t | y_{<t}, x, x_v) = p_\theta(y_t | Q_t, \hat{K}_t, \hat{V}_t)$. By selectively emphasizing tokens that carry contextual information, it ensures that the model maintains consistent alignment with the visual content while reducing the sequence length.

## 4.2 COLLABORATIVE DECODING WITH ORIGINAL DECODING STRATEGY

Relying solely on image-focused decoding result in the model failing to fully capture detailed information. The detailed experiment of this issue can be found in Section 5.2. To address this concern, we propose combining the original inference decoding with a shorter sequence decoding

that is more focused on the image. This approach is expected to enhance decoding while maintaining the stability of the inference process.

Building on the key-value merging discussed in Section 4.1, we derive the following equation:

$$p(y_t|y_{<t}, x, x_v) = p_\theta(y_t|Q_t, K_t, V_t) + \alpha p_\theta(y_t|Q_t, \hat{K}_t, \hat{V}_t), \tag{9}$$

where $\alpha$ is a hyper-parameter that balances the original inference decoding with the image-focused decoding. By effectively leveraging both the standard and image-focused decoding strategies, our method seeks to improve the model's performance. This integration of key-value merging with adaptive decoding represents a significant step towards more image-conditioned language generation.

### 4.3 ADAPTIVE PLAUSIBILITY CONSTRAINTS

Though collaborative decoding based on image attention enhance the LVLMs' alignment, there still exists a challenge. The logits of some implausible tokens may be unexpectedly enhanced. Those tokens with very low confidence are commonly implausible or hallucinated, not grounded in images. Through image-guided key-value merging, these logits with low confidence may be enhanced as well, affecting the performance of LVLMs. To tackle this issue, we draw inspiration from previous works (Li et al., 2022; Leng et al., 2024) and adopt an adaptive plausibility constraint for our method. Specifically, we select next token from those tokens whose probabilities exceed a predefined confidence level in the original output distribution, denoted as follows:

$$\mathcal{V}_{\text{head}}\ (y_{<t}) = \{y_t \in \mathcal{V} : p(y_t|y_{<t}, x, x_v) \geq \beta \max_w p(w|y_{<t}, x, x_v)\},$$
$$p(y_t|y_{<t}, x, x_v) = 0, \text{ if } y_t \notin \mathcal{V}_{\text{head}}\ (y_{<t}), \tag{10}$$

where $\mathcal{V}$ is the output vocabulary of LVLM and $\beta$ is a hyper-parameter between 0 and 1 to control the truncation of the next token distribution. A larger $\beta$ means a more strict restriction to the selection of next token, retaining only high-probability tokens.

## 5 EXPERIMENT

In this section, we evaluate IKOD in aligning vision and language modalities in LVLMs and improving the model performance. We aim to answer the following questions: (1) Can IKOD reduce hallucination in LVLMs? (2) How does IKOD improve model performance in comprehensive benchmarks? (3) Does the key component of IKOD contribute to the model's performance?

### 5.1 EXPERIMENTAL SETTINGS

**Evaluation Benchmarks.** We conduct evaluations on both hallucination benchmarks and comprehensive benchmarks. Specially, this includes: (1) Hallucination benchmarks (POPE (Li et al., 2023b), CHAIR (Rohrbach et al., 2018)). (2) Comprehensive benchmarks (VQAv2 (Goyal et al., 2017), ScienceQA (SQA) (Lu et al., 2022), MME (Fu et al., 2024), MMBench (Liu et al., 2023c), MM-Vet (Yu et al., 2023b), COCO Caption (Chen et al., 2015)). More details are provided in Appendix A.3.

**Baselines.** First, We compare our approach to existing decoding methods: Nucleus sampling (p = 0.1), Greedy search, OPERA (Huang et al., 2023), VCD (Leng et al., 2024), HALC (Chen et al., 2024b) and AGLA (An et al., 2024). Furthermore, We compare the performance of IKOD with other LVLM preference tuning methods, including Silkie (Li et al., 2023a), LLaVA-RLHF (Sun et al., 2023), and RLHF-V (Yu et al., 2024b). More details about these methods can be found in Appendix A.4.

**Implementation Details.** Following previous research (An et al., 2024; Leng et al., 2024), We utilize LLaVA-1.5 (Liu et al., 2024b) and InstructBLIP (Dai et al., 2023) with the language decoder Vicuna 7B as the backbone models. In all experiments unless specially mentioned, we adopt Greedy search as the base decoding strategy for IKOD and other methods. The comprehensive pararmeter settings are detailed in Appendix A.5. For compared methods, we follow the suggested settings in their respective papers and released codes to ensure a fair comparison, and the random seed is fixed to 100 coherently. All experiments are conducted on a single NVIDIA A100 GPU.

Table 1: F1 score on POPE-MSCOCO dataset. We **Bold** the best results and underline the second best results.

| Model | Decoding | Random | Popular | Adversarial | Average |
|---|---|---|---|---|---|
| LLaVA1.5 | Nucleus | 81.07 | 80.30 | 77.81 | 79.73 |
| | Greedy | 85.50 | 84.37 | 82.32 | 84.06 |
| | OPERA | 84.52 | 85.38 | 81.51 | 83.20 |
| | VCD | 87.91 | 85.83 | 82.16 | 85.30 |
| | HALC | 84.48 | 83.53 | 81.51 | 83.17 |
| | AGLA | 86.32 | 85.21 | **83.27** | 84.93 |
| | **IKOD** | **89.88** | **87.86** | 83.11 | **86.95** |
| InstructBLIP | Nucleus | 81.13 | 78.75 | 77.83 | 79.24 |
| | Greedy | 86.98 | 84.31 | 82.13 | 84.47 |
| | OPERA | 87.12 | 82.22 | 80.73 | 84.54 |
| | VCD | 85.72 | 83.21 | 81.24 | 83.39 |
| | HALC | 87.05 | 84.29 | 82.17 | 84.50 |
| | AGLA | 87.00 | 84.35 | 81.86 | 84.40 |
| | **IKOD** | **87.57** | **85.15** | **82.46** | **85.06** |

Table 2: Evaluation results on COCO caption benchmark. Lower $CHAIR_S$ and $CHAIR_I$ indicate fewer hallucinations, and higher recall and BLEU-4 indicate better performance.

| Model | Decoding | $CHAIR_S \downarrow$ | $CHAIR_I \downarrow$ | Recall $\uparrow$ | BLEU-4 $\uparrow$ | Avg. Len |
|---|---|---|---|---|---|---|
| LLaVA-1.5 | Nucleus | 57.2 | 14.6 | 76.5 | 3.1 | 105.6 |
| | Greedy | 50.0 | 12.0 | 81.9 | 4.8 | 101.0 |
| | OPERA | 48.6 | 11.2 | **82.6** | 4.9 | 95.2 |
| | VCD | 50.8 | 11.8 | 81.1 | 4.5 | 100.9 |
| | HALC | 40.2 | **8.1** | 77.1 | 5.0 | 94.2 |
| | AGLA | 50.0 | 12.1 | 81.9 | 4.8 | 100.6 |
| | **IKOD** | **36.4** | 8.8 | 80.9 | **5.2** | 99.5 |
| InstructBLIP | Nucleus | 57.6 | 14.8 | 71.9 | 2.8 | 111.1 |
| | Greedy | 46.2 | 10.4 | 76.4 | 4.9 | 102.4 |
| | OPERA | 50.6 | 12.6 | 75.9 | 0.8 | 97.3 |
| | VCD | 52.4 | 12.2 | 76.8 | 4.9 | 98.6 |
| | HALC | 60.2 | 18.0 | 74.8 | 3.9 | 106.0 |
| | AGLA | 46.4 | 10.4 | 76.5 | **5.0** | 102.4 |
| | **IKOD** | **39.8** | **6.9** | **78.8** | 4.6 | 119.2 |

## 5.2 EXPERIMENTAL RESULTS

**Results on POPE.** The text instruction we used for POPE is "Is there object in this the image? Please answer this question with one word." Table 1 presents the results on POPE-MSCOCO dataset (Li et al., 2023b) across various baselines and backbone models. The F1 scores are reported for three distinct task types: Random, Popular, and Adversarial. Notably, significant improvements are observed when comparing IKOD with other methods, thereby underscoring its efficacy in enhancing the performance of LVLMs.

**Results on CHAIR.** In the CHAIR benchmark, we randomly select 500 images from MSCOCO validation dataset (Lin et al., 2014) to conduct an evaluation. We adopt "Please describe this image in detail." as the text instruction. The results compared with other methods are presented in Table 2. Obviously, IKOD outperforms other approaches on $CHAIR_S$ and $CHAIR_I$ metrics significantly. In BLEU-4 scores and recall scores, IKOD achieve superior performance, effectively improving the accuracy of the generated captions. Moreover, IKOD does not shorten the generated sequence length, demonstrating its ability to preserve diversity in the output. This comparasion indicates that IKOD effectively mitigate hallucinations and improve modality alignment in LVLMs.

**Results on Comprehensive Benchmark.** We provide a comprehensive benchmark comparison between IKOD and other approaches, as illustrated in Table 3 and Table 4. Despite the varied strategies used for different LVLMs, IKOD consistently outperforms other LVLMs in comprehensive benchmarks. This comparison underscores IKOD's exceptional ability to integrate image and text modalities, leading to an enhancement in LVLMs' performance. To have a detailed comparison,

we evaluate the perception and cognition ability of IKOD and other decoding methods on MME benchmark, where IKOD has a better performance as well. Details are shown in Appendix A.7

Table 3: The performance of adopting IKOD on LLaVA-1.5 across comprehensive benchmarks.

| Method | VQA$^{v2}$ ↑ | SQA$^{I}$ ↑ | VQA$^{T}$ ↑ | MME ↑ | MMBench ↑ | MM-Vet ↑ | COCO-caption ↑ |
|---|---|---|---|---|---|---|---|
| LLaVA-1.5 | 76.5 | 66.8 | 46.0 | 1458.8 | 64.3 | 30.5 | 56.6 |
| + IKOD | **76.7** | **68.1** | **46.1** | **1489.4** | **64.4** | **31.1** | **56.8** |

**Ablation Studies - KV merging Strategy**. In section 4.1, we select the tokens with lower attention on image in text sequence as anchors while merging other tokens' Keys and Values into the anchors'. To verify its effectiveness, we conduct an ablation study and compare the performance of three KV merging strategies. Specially, we randomly selecting tokens (Random), selecting high attention tokens (High Attention), selecting low attention tokens (Low Attention (Ours)) as anchors, and other tokens' KVs are merged. The comparison results are presented in Table 5. It's obvious that our method, namely selecting low attention tokens as anchors, has the best performance across all ratios. This is reasonable as the tokens with low attention on image are commonly appears at the end of the sequence, which are more relevant with the last token namely query token. Retaining these tokens and merging other tokens can reserve more contextual information and get a shorter sequence with higher attention on image, contributing to generating more rational text grounded in image.

Table 4: Comparison between IKOD and other preference construction approaches across hallucination and comprehensive evaluation benchmarks.

| **Metric** | LLaVA-1.5 | + Vlfeedback | + Human-Preference | + RLHF-V | + **IKOD** |
|---|---|---|---|---|---|
| CHAIR$_S$ ↓ | 45.0 | 43.6 | 44.0 | 44.6 | **36.4** |
| CHAIR$_i$ ↓ | 10.1 | 9.4 | 9.3 | **7.9** | 8.8 |
| POPE ↑ | 85.9 | 83.7 | 81.5 | 86.2 | **87.0** |
| SciQA-IMG ↑ | 66.8 | 66.2 | 65.8 | 67.1 | **68.1** |
| MM-Vet ↑ | 30.5 | **31.2** | 31.1 | 30.9 | 31.1 |
| MMBench ↑ | 63.0 | 63.9 | 60.4 | 63.6 | **64.4** |
| MME ↑ | 1458.8 | 1432.7 | 1490.6 | **1498.3** | 1489.4 |

Table 5: F1 Score comparison of different KV merging strategies across various anchor ratios on POPE-MSCOCO dataset under random setting.

| KV Merging Strategies | 0.9 | 0.8 | 0.7 | 0.6 | 0.5 | 0.4 | 0.3 | 0.2 | 0.1 |
|---|---|---|---|---|---|---|---|---|---|
| Random | 86.12 | 86.84 | 85.31 | 84.64 | 83.44 | 83.36 | 83.38 | 81.53 | 79.73 |
| High Attention | 77.57 | 81.33 | 84.62 | 83.86 | 82.58 | 83.05 | 83.51 | 84.79 | 85.42 |
| **Low Attention(Ours)** | **87.68** | **88.37** | **86.68** | **87.38** | **88.83** | **89.88** | **88.35** | **87.60** | **78.44** |

**Effect of Anchor Ratio** $\lambda$. The anchor ratio $\lambda$ is an important hyper-parameter reflecting the degree of KV Cache compression. A higher $\lambda$ indicates more tokens are reserved and a lower degree of KV Cache compression, and $\lambda = 1$ implies the original generation procedure with full cache. We conduct an analysis on POPE-MSCOCO dataset to explore its effect. The results are depicted in Figure 6. We can easily draw the conclusion that when $\lambda$ is too small or too big the model's performance are restricted, and $\lambda = 0.4$ is the best anchor ratio for both LLaVA-1.5 and InstructBLIP. The explanation for this phenomenon could be summarized into two points: (1) Lower $\lambda$ means less tokens are selected as anchors, along with an excessive compressed KV Cache, resulting in a significant information loss which is adverse to the next token generation. (2) Higher $\lambda$ means the tokens with low attention on image are not augmented enough by KV Cache compression. Based on the analysis, we set $\lambda$ to 0.4 unless sepecially stated to get a better performance. More ablation studies and case studies can be found in Appendix A.8 and Appendix A.9 respectively.

## 6 RELATED WORK

### 6.1 LARGE VISION-LANGUAGE MODELS

In recent years, significant advancements in Large Language Models (LLMs) (Brown et al., 2020; OpenAI, 2023; Touvron et al., 2023) have fueled the development of Large Vision-Language Models

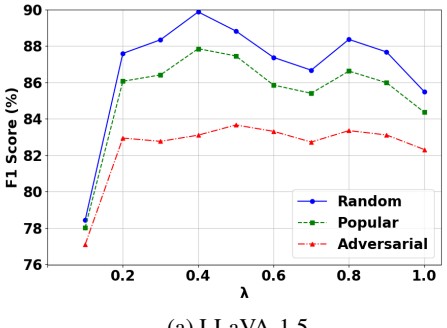 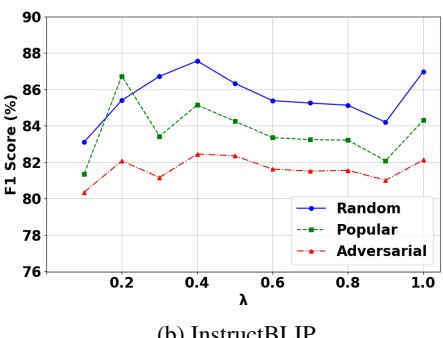

(a) LLaVA-1.5                                  (b) InstructBLIP

Figure 6: IKOD performance on POPE-MSCOCO dataset across different anchor ratios $\lambda$ on LLaVA-1.5 and InstructBLIP.

(LVLMs). These models effectively integrate large-scale pre-trained vision models into the LLMs' representation space. LVLMs are generally classified into two main types: MLP-based models and Q-former-based models. These models have demonstrated strong performance by combining LLMs with image inputs, achieving notable success in image comprehension tasks. However, despite these successes, LVLMs are not without flaws. They often encounter issues like "hallucinations," where the generated outputs fail to accurately reflect the content of the input image.

To address these challenges, recent studies have focused on methods such as instruction tuning (Lin et al., 2023; Dai et al., 2024; Liu et al., 2024c), post-processing (Zhou et al., 2023; Yin et al., 2023), preference tuning (Yu et al., 2023a; Zhou et al., 2024), and decoding strategies (Huang et al., 2023; Chen et al., 2024b) to enhance the alignment between visual and textual information. However, these approaches often come with significant drawbacks. Instruction tuning and preference tuning methods require costly dataset annotation, introduce unintended biases, and demand extensive computational resources. Post-processing solutions correct hallucinated tokens in real-time, often relying on external tools like pre-trained vision-language models and stronger foundational models.

## 6.2 DECODING STRATEGIES FOR LVLMS

Decoding strategies are crucial for large models, as they determine how the model generates corresponding responses based on images and instructions. Additionally, they can enhance model performance without the need for training. They play a pivotal role in shaping the output's quality, relevance, and coherence. Traditional strategies such as greedy decoding, nucleus sampling, beam search, provide a variety of options for large models in terms of output diversity, reliability, and certainty balance between randomness and relevance. Recently, decoding strategies for large foundation models have primarily concentrated on contrasting logits across different layers (Chuang et al., 2023), applying logit penalties (Huang et al., 2023), and employing contrastive decoding (Leng et al., 2023; Chen et al., 2024b).

## 7 CONCLUSION

In this paper, we investigate the impact of sequence length on image attention in Large Vision-Language Models (LVLMs), specifically focusing on how attention weakens as the sequence progresses. Our analysis revealed a significant reduction in image attention towards the end of sequences, which correlates with a higher occurrence of hallucinated tokens and performance degradation in the model. To address this issue, we introduce an image attention-guided Key-Value Merging strategy, designed to enhance the model's focus on visual elements by selectively merging key and value vectors based on their attention scores. Furthermore, we propose a collaborative decoding method named IKOD that combines the logits derived from the compressed KV Cache and original logits to obtain a output distribution more grounded in image. Our experiments demonstrate that IKOD can not only mitigate hallucinations in LVLMs but also enhance their comprehensive capacities, dismissing the need for additional training or external tools and making it applicable to various models.

## IMPACT STATEMENT

This paper presents work whose goal is to advance the field of Machine Learning. There are many potential societal consequences of our work, none which we feel must be specifically highlighted here.

## REPRODUCIBILITY STATEMENT

For our empirical results, we provide a comprehensive overview of baseline details delve into the details of the experimental sttings, all of which can be found in Section 5.1, Appendices A.3, A.4 and A.5. Additionally, in Appendix A.9, we offer detailed case demonstrations and comparisons. It is worth noting that we are committed to open source the code related to our research after publication.

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

# A  APPENDIX

## A.1  LIMITATIONS

Though there are many strengths for IKOD, we still acknowledge that it has some limitations. As we obtain an augmented view on input image through key-value merging, it's not always beneficial in some cases. When the input image has some misleading information, excessive focus on the image could make models prone to generating responses that go against common sense. Moreover, the hyper-parameter $\alpha$ modulating the balance of augmented and original output distributions and the anchor ratio $\lambda$ controlling the degree of KV Cache compression need to be set manually, which limits its convenience to some extent. In the future study we will try to explore self-adaptive methods to substitute them.

## A.2  IMPLEMENTATION ABOUT THE VISUALIZATION

### A.2.1  DETAILS ABOUT THE VISUALIZATION METRIC

KDE is a non-parametric way to estimate the probability density function of a random variable by smoothing out the data points. The idea behind KDE is to estimate the distribution of data points by placing a kernel function on each data point and summing them up to create a smooth estimate of the data's probability density. For two-dimensional data $x$ and $y$, the KDE is defined by the following formula:

$$\hat{f}(x,y) = \frac{1}{nh_x h_y} \sum_{i=1}^{n} K\left(\frac{x-x_i}{h_x}, \frac{y-y_i}{h_y}\right), \tag{11}$$

where:

- $\hat{f}(x, y)$ is the estimated density at the point $(x, y)$.

- $n$ is the number of data points.

- $K(\cdot)$ is the kernel function, typically a Gaussian kernel:

$$K(u, v) = \frac{1}{2\pi} e^{-\frac{1}{2}(u^2 + v^2)}$$

- $h_x$ and $h_y$ are the bandwidth parameters, which control the smoothness of the density estimate. We set both $h_x$ and $h_y$ to 0.5 in our analysis.

### A.2.2 MORE EXAMPLES OF VISUALIZATION OF ATTENTION IN LVLMs

We conduct an analysis on the relationship between image attention and token position across different Large Vision-Language Models (LVLMs), as well as the relationship between image attention and model performance. We present the visualization in Figure 7. A similar phenomenon IS observed across different models: as the sequence length increases, image attention diminishes, particularly towards tokens appearing later in the sequence. Also we find that weakened attention is correlated with a higher concentration of hallucinated tokens in areas with low attention, indicating that reducing image attention is more likely to lead to errors in the model.

### A.3 EVLAUATION METRICS AND BENCHMARKS

**POPE.** The Polling-based Object Probing Evaluation (POPE) (Li et al., 2023b) is a widely-used benchmark to assess object halucination in LVLMs, which contains 27,000 Yes/No questions in three datasets: MSCOCO (Lin et al., 2014), A-OKVQA (Schwenk et al., 2022), GQA (Hudson & Manning, 2019). Each dataset has three nagative sample settings: random, popular, adversarial. It adpots Accuracy, Precision, Recall, and F1 score as the evaluation metrics.

**CHAIR.** Caption Hallucination Assessment with Image Relevance (CHAIR) (Rohrbach et al., 2018) is a popular method to evaluate object hallucination in image caption tasks. It compares generated objects with grounde-truth objects to calculate the degree of hallucination. CHAIR evaluate object hallucination from two dimensions: instance-level and sentence-level, denoted as $\text{CHAIR}_I$ and $\text{CHAIR}_S$ respectively, which are computed as:

$$\text{CHAIR}_I = \frac{|\{\text{hallucinated objects}\}|}{|\{\text{all mentioned objects}\}|} \quad \text{CHAIR}_S = \frac{|\{\text{captions with hallucinated objects}\}|}{|\{\text{all captions}\}|}$$

**VQAv2.** VQAv2 (Goyal et al., 2017) balances the popular VQAdataset (Antol et al., 2015) by collecting complementary images such that every question in the balanced dataset is associated with a pair of similar images that result in two different answers to the question. It has approximately twice the number of image-question pairs.

**SQA.** ScienceQA (SQA) (Lu et al., 2022) is a benchmark that consists of 21k multimodal multiple choice questions within the domain of science, along with annotations of their answers and corresponding lectures and explanations.

**MME.** Multimodal Large Language Model Evaluation (MME) (Fu et al., 2024) is a comprehensive benchmark to assess the capabilities of LVLMs in multimodal tasks. It evaluates models with the total score of Accuracy and Accuracy+ across two primary dimensions: perception and cognition, containing 10 and 4 meticulously designed subtasks respectively.

**MMBench.** MMBench (Liu et al., 2023c) is a meticulously curated dataset expanding the scope of evaluation questions and abilities. It introduces a rigorous CircularEval strategy which leverages large language models to convert free-form predictions into pre-defined choices, resulting in more accurate evaluation results.

**MM-Vet.** MM-Vet (Yu et al., 2023b) is an evaluation benchmark to assess the performance of LVLMs on complicated multimodal tasks, which focus on six core vision-language capabilities: recognition, knowledge, optical character recognition (OCR), spatial awareness, language generation, and math.

**COCO Caption.** The Microsoft COCO Caption dataset (Chen et al., 2015) contains over one and a half million captions corresponding to more than 330,000 images. It used an evaluation server to score candidate captions using popular metrics, including BLEU, METEOR, ROUGE and CIDEr.

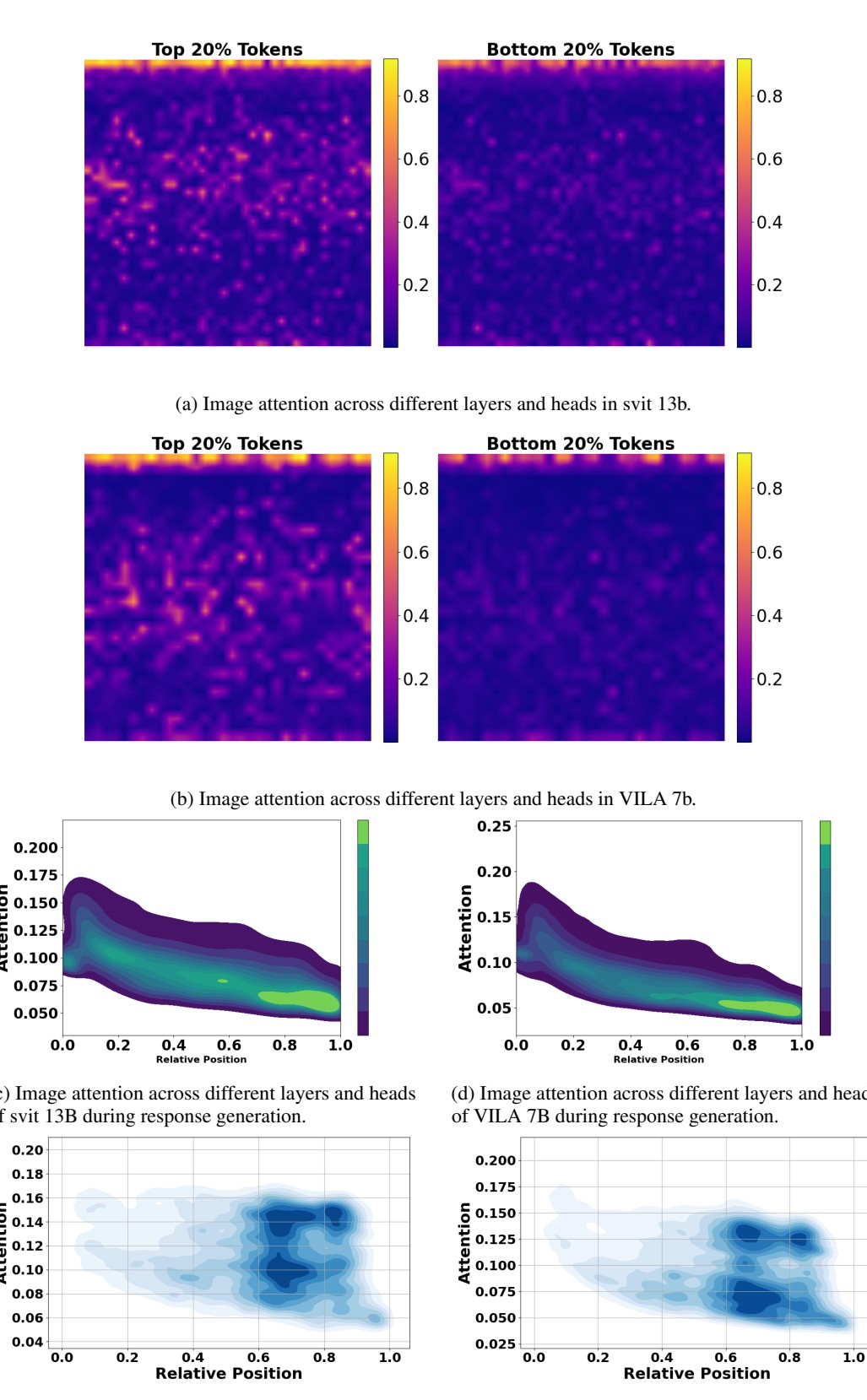

(a) Image attention across different layers and heads in svit 13b.

(b) Image attention across different layers and heads in VILA 7b.

(c) Image attention across different layers and heads of svit 13B during response generation.

(d) Image attention across different layers and heads of VILA 7B during response generation.

(e) Relationship between image attention and model performance on svit.

(f) Relationship between image attention and model performance on VILA.

Figure 7: More examples of attention visualization.

## A.4 OVERVIEW OF THE BASELINES

**LLaVA-1.5.** LLaVA-1.5 (Liu et al., 2024b) is an improvement based on LLaVA (Liu et al., 2024c). It modifies with a CLIP-ViT-L-336px visual backbone and MLP projection and incorporates academic-task-oriented VQA data with response formatting prompts, achieving state-of-the-art across 11 benchmarks at that time.

**InstructBLIP.** InstructBLIP (Dai et al., 2023) utilizes an instruction-aware Query Transformer to extracts informative features tailored to the given instruction, demonstrating significant instruction following ability. It achieves state-of-the-art zero-shot performance across 13 datasets and also excels in some finetuned downstream tasks, like ScienceQA.

**OPERA.** OPERA (Huang et al., 2023) is a novel MLLM decoding method based on an Over-trust Penalty and a Retrospection-Allocation strategy. It adds a penalty to the model logits to mitigate the over-trust issue on summary token, along with a rollback strategy to correct the token selection.

**VCD.** Visual Contrastive Decoding (VCD) (Leng et al., 2024) calibrates model's outputs through contrasting output distributions derived from original and distorted visual inputs, thus reducing the the over-reliance on statistical bias and unimodal priors, significantly mitigating the object hallucination issue across different LVLMs.

**HALC.** HALC (Chen et al., 2024b) is a plug-and-play decoding algorithm to mitigate object hallucination in LVLMs. It operates on both local and global contexts, integrating a robust auto-focal grounding mechanism to correct hallucinated tokens and a specialized beam search algorithm promoting further visually matched generations.

**AGLA.** AGLA (An et al., 2024) leverages an image-prompt matching scheme to get an augmented view of the input image where prompt-relevant content is reserved while others are masked. With the augmented view, models can calibrate the output distribution by integrating generative global features and discriminative local features.

**Silkie.** Silkie (Li et al., 2023a) utilizes AI annotation to build a vision-language feedback (VLFeed-back) dataset. With preference distillation through direct preference optimization (DPO) on it, Silkie achieves more comprehensive improvements compared to human-annotated preference datasets.

**LLaVA-RLHF.** LLaVA-RLHF (Sun et al., 2023) introduces Reinforcement Learning from Human Feedback (RLHF) from the text domain to the task of vision-language alignment. With the propsed Factually Augmented RLHF, it augments the reward model with additional factual information and alleviates the reward hacking phenomenon in RLHF, resulting in a performance improvement.

**RLHF-V.** RLHF-V (Yu et al., 2024b) collects human preference on segment-level and performance dense direct preference optimization on it, achieveing state-of-the-art performance in trustworthiness among open-source LVLMs at that time.

## A.5 EXPERIMENTAL SETTINGS

In all experimental setups, we fix anchor ratio $\lambda$ to 0.4 and $\beta$ to 0.1 unless explicitly stated otherwise. For POPE and CHAIR, We set $\alpha$ to 2 for LLaVA-1.5, while setting $\alpha$ to 1.1 for InstructBLIP. For MME, $\alpha$ is set to 0.8, and $\lambda$ is set to 0.9 and 0.8 for LLaVA-1.5 and InstructBLIP respectively. For other benchmarks, the hyper-parameters are the same as POPE's on LLaVA-1.5.

## A.6 POPE EXPERIMENT DETAILS

We show the full results on POPE-MSCOCO dataset in Table 6. From the table, we can see that the proposed decoding strategy IKOD consistently outperforms other methods in terms of accuracy and F1 Score across nearly all settings, especially on random setting, demonstrating the significant strength of our method. Though we don't achieve the best performance on adversarial setting, which may be attributed to the frequent co-occurence schemes in pretrained datasets and our excessive attention on image, IKOD still gains the suboptimal results, proving its superiority.

Table 6: POPE results on MSCOCO dataset. Higher accuracy and F1 score indicate better performance. **Bold** indicates the best results of all methods.

| Setting | Model | Decoding | Accuracy | Precision | Recall | F1 Score |
|---------|-------|----------|----------|-----------|--------|----------|
| *Random* | LLaVA-1.5 | Nucleus | 82.97 | 91.24 | 72.93 | 81.07 |
| | | Greedy | 87.07 | 97.28 | 76.27 | 85.50 |
| | | OPERA | 86.30 | 97.14 | 74.80 | 84.52 |
| | | VCD | 88.37 | 91.49 | 84.60 | 87.91 |
| | | HALC | 86.27 | 97.14 | 74.73 | 84.48 |
| | | AGLA | 87.73 | 97.56 | 77.40 | 86.32 |
| | | **Ours** | **90.17** | 92.58 | 87.33 | **89.88** |
| | InstructBLIP | Nucleus | 81.37 | 82.07 | 80.27 | 81.16 |
| | | Greedy | 87.97 | 94.81 | 80.33 | 86.97 |
| | | OPERA | 88.07 | 94.61 | 80.73 | 87.12 |
| | | VCD | 86.77 | 93.05 | 79.47 | 85.72 |
| | | HALC | 88.03 | 94.82 | 80.47 | 87.05 |
| | | AGLA | 88.00 | 94.88 | 80.33 | 87.00 |
| | | **Ours** | **88.23** | 92.77 | 82.93 | **87.57** |
| *Popular* | LLaVA-1.5 | Nucleus | 82.10 | 89.31 | 72.93 | 80.30 |
| | | Greedy | 85.87 | 84.39 | 76.27 | 84.37 |
| | | OPERA | 85.30 | 94.68 | 74.80 | 85.38 |
| | | VCD | 86.03 | 87.10 | 84.60 | 85.83 |
| | | HALC | 85.27 | 94.68 | 74.73 | 83.53 |
| | | AGLA | 86.57 | 94.78 | 77.40 | 85.21 |
| | | **Ours** | **87.93** | 88.39 | 87.33 | **87.86** |
| | InstructBLIP | Nucleus | 79.23 | 78.46 | 80.60 | 79.51 |
| | | Greedy | 85.00 | 88.60 | 80.33 | 84.27 |
| | | OPERA | 84.93 | 88.14 | 80.73 | 84.27 |
| | | VCD | 83.97 | 87.33 | 79.47 | 83.21 |
| | | HALC | 85.00 | 88.49 | 80.47 | 84.29 |
| | | AGLA | 85.10 | 88.80 | 80.33 | 84.35 |
| | | **Ours** | **85.53** | 87.48 | 82.93 | **85.15** |
| *Adversarial* | LLaVA-1.5 | Nucleus | 79.20 | 83.38 | 72.93 | 77.81 |
| | | Greedy | 83.63 | 89.51 | 76.20 | 82.32 |
| | | OPERA | 83.07 | 89.74 | 74.67 | 81.51 |
| | | VCD | 81.63 | 79.86 | 84.60 | 82.16 |
| | | HALC | 83.07 | 89.81 | 74.60 | 81.51 |
| | | AGLA | **84.47** | 90.20 | 77.33 | **83.27** |
| | | **Ours** | 82.27 | 79.33 | 87.27 | 83.11 |
| | InstructBLIP | Nucleus | 77.40 | 76.08 | 79.93 | 77.96 |
| | | Greedy | 82.47 | 83.77 | 80.53 | 82.12 |
| | | OPERA | **82.51** | 83.55 | 80.93 | 82.22 |
| | | VCD | 81.63 | 83.02 | 79.53 | 81.24 |
| | | HALC | 82.50 | 83.74 | 80.67 | 82.17 |
| | | AGLA | 82.17 | 83.30 | 80.47 | 81.86 |
| | | **Ours** | 82.33 | 81.87 | 83.07 | **82.46** |

## A.7 MME EXPERIMENT DETAILS

To compare the performance of IKOD and other decoding methods, we conduct comprehensive experiments on MME benchmark based on the backbones of LLaVA-1.5 and InstructBLIP. As illustrated in Table 7 and 8, our method achieve the best performance on perception capability and suboptimal results on cognition capability for LLaVA-1.5. For InstructBLIP, despite IKOD lags behind VCD on perception capability, it surpasses all other methods on cognition capability, further demonstrate IKOD can improve LVLMs' comprehensive capacities. As for the subtasks, each method has its own advantages, so we don't make a specific comparison.

Table 7: Results on MME perception-related tasks.

| Model | Decoding | *Existence* | *Count* | *Position* | *Color* | *Posters* | *Celebrity* | *Scene* | *Landmark* | *Artwork* | *OCR* | *Perception Total* |
|---|---|---|---|---|---|---|---|---|---|---|---|---|
| | Nucleus | 180.00 | 101.67 | 111.67 | 140.00 | 105.10 | 111.76 | 144.50 | 122.50 | 101.75 | 100.00 | 1218.95 |
| | Greedy | 195.00 | 158.33 | 123.33 | 155.00 | 129.59 | 133.53 | 154.75 | 163.25 | 121.00 | 125.00 | 1458.79 |
| LLaVA-1.5 | VCD | 185.00 | 153.33 | **133.33** | 138.33 | 130.27 | **152.94** | 148.25 | **166.00** | 123.50 | 130.00 | 1460.96 |
| | AGLA | 195.00 | 155.00 | 133.33 | 160.00 | **142.86** | 133.53 | 156.25 | 164.50 | 114.50 | 132.50 | 1487.47 |
| | Ours | **195.00** | **173.33** | 128.33 | **160.00** | 129.59 | 137.65 | **156.50** | 159.25 | 117.25 | **132.50** | **1489.41** |
| | Nucleus | 168.33 | 51.67 | **56.67** | 115.00 | 117.01 | 97.65 | 147.00 | 132.75 | 92.75 | 80.00 | 1058.82 |
| | Greedy | 185.00 | **60.00** | 50.00 | 120.00 | 141.84 | 80.00 | 160.00 | 159.25 | 91.50 | 65.00 | 1112.59 |
| InstructBLIP | VCD | 185.00 | 60.00 | 51.67 | **123.33** | 150.68 | **97.65** | 156.50 | 161.50 | 96.00 | **102.50** | **1184.83** |
| | AGLA | 185.00 | 60.00 | 50.00 | 120.00 | 141.84 | 82.65 | **160.50** | 160.00 | 91.50 | 65.00 | 1116.48 |
| | Ours | **185.00** | 55.00 | 48.33 | 105.00 | **156.80** | 92.35 | 159.50 | 154.25 | 89.25 | 87.50 | 1132.99 |

Table 8: Results on MME cognition-related tasks.

| Model | Decoding | *Common Sense Reasoning* | *Numerical Calculation* | *Text Translation* | *Code Reasoning* | *Cognition Total* |
|---|---|---|---|---|---|---|
| | Nucleus | 107.86 | **60.00** | 57.50 | **97.50** | **322.86** |
| | Greedy | **120.71** | 50.00 | 50.00 | 77.50 | 298.21 |
| LLaVA-1.5 | VCD | 120.71 | 47.50 | 57.50 | 72.50 | 298.21 |
| | AGLA | 115.00 | 37.50 | 50.00 | 62.50 | 265.00 |
| | Ours | 120.00 | 55.00 | **57.50** | 67.50 | 300.00 |
| | Nucleus | 72.86 | **90.00** | 50.00 | 40.00 | 252.86 |
| | Greedy | 97.86 | 47.50 | 50.00 | 45.00 | 240.36 |
| InstructBLIP | VCD | **102.14** | 45.00 | **57.50** | **47.50** | 252.14 |
| | AGLA | 97.86 | 47.50 | 50.00 | 45.00 | 240.36 |
| | Ours | 99.29 | 42.50 | **70.00** | 45.00 | **256.79** |

## A.8 ABLATION STUDIES

### A.8.1 EFFECT OF $\alpha$

$\alpha$ is an important hyper-parameter which modulates the level of amplification between original and augmented output distributions, as formulated in Equation 9. Figure 8 demonstrates the outcomes of an ablation study focusing on $\alpha$, from where we can observe the trend of model's performance increasing first and then decreasing as $\alpha$ grows, and the best $\alpha$ are 2 and 1.1 for LLaVA-1.5 and InstructBLIP respectively. When $\alpha$ is small, the effect of amplification is not obvious. Conversely, too large $\alpha$ could break the balance of original and augmented output distribution, distorting model's inherent parameter information.

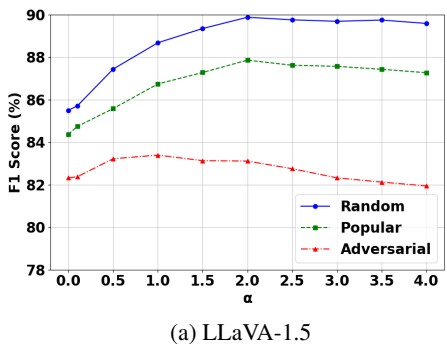
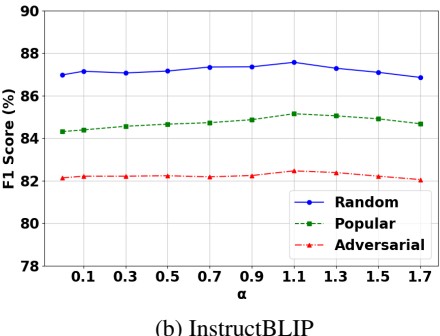

| (a) LLaVA-1.5 | (b) InstructBLIP |
|---|---|

Figure 8: IKOD performance on POPE-MSCOCO dataset across different $\alpha$ on LLaVA-1.5 and InstructBLIP.

### A.8.2 EFFECT OF $\beta$

$\beta$ controls the adaptive plausible constraint in Equation 10. As the constraint is set based on the max logit of candidate tokens, it may not work for greedy decoding. So we adopt nucleus sampling (p = 0.1) to explore the effect of $\beta$. The ablation results are shown in Figure 9. $\beta = 0$, implying no constraint, has suboptimal performance, validating our rationale for implementing this constraint.

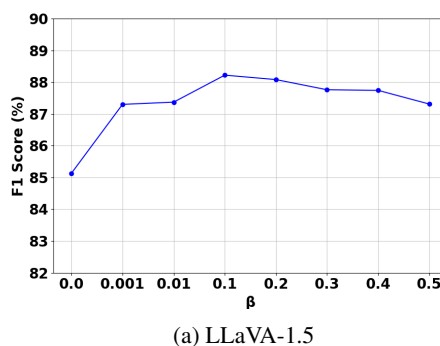 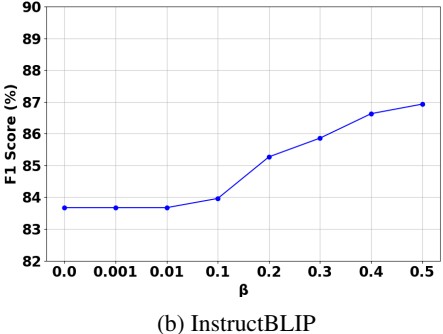

(a) LLaVA-1.5                (b) InstructBLIP

Figure 9: IKOD performance on POPE-MSCOCO dataset under the random setting across different $\beta$ on LLaVA-1.5 and InstructBLIP.

For LLaVA-1.5, F1 score increases first and then decreases as $\beta$ increases, while for InstructBLIP, F1 score grows continuously, indicating that the best threshold for the constraint is low for LLaVA-1.5 and high for InstructBLIP. Too large $\beta$ may exclude the valid tokens unexpectedly. When applied, we encourage users to set it to a rational value, like 0.1.

### A.8.3 EFFECT OF DIFFERENT SAMPLING STRATEGIES

Following VCD's setting (Leng et al., 2024), we conduct an ablation study on various sampling strategies using POPE-MSCOCO dataset under the random setting with LLaVA-1.5 backbone. In addition to the greedy search approach discussed in the main paper, this experiment includes four additional sampling strategies: Top P sampling (specifically, p = 0.9), Top K sampling (specifically, k = 50), Nucleus s, and Top K sampling with temperature normalization (k = 50, temp = 1.5/0.7). Results are presented in Table 9. We can observe that applying IKOD, irrespective of the sampling strategy employed, consistently contributes to hallucination mitigation in LVLMs. This consistency underscores the versatility and effectiveness of IKOD in enhancing the alignment of vision and language in LVLMs.

Table 9: An ablation study of different sampling strategies.

| Sampling Strategy | w. IKOD | Accuracy | Precision | Recall | F1 Score |
|---|---|---|---|---|---|
| Top P | No | 86.63 | 96.14 | 76.33 | 85.10 |
| | Yes | **89.60** | 91.17 | 87.07 | **89.33** |
| Top K | No | 82.97 | 91.24 | 72.93 | 81.07 |
| | Yes | **88.53** | 90.99 | 85.53 | 88.18 |
| Nucleus | No | 82.97 | 91.24 | 72.93 | 81.07 |
| | Yes | **88.57** | 91.00 | 85.60 | **88.22** |
| Top K+Temperature 0.7 | No | 86.93 | 96.63 | 76.53 | 85.42 |
| | Yes | **89.97** | 92.07 | 87.47 | 89.71 |
| Top K+Temperature 1.5 | No | 86.27 | 95.26 | 76.33 | 84.75 |
| | Yes | **89.47** | 91.75 | 86.73 | **89.17** |

### A.9 CASE STUDIES

We leverage CHAIR evaluation as a case study to qualitatively compare the decoding outputs of IKOD with other methods. Responses generated by IKOD and other decoding methods including OPERA, VCD, HALC and AGLA incorporating LLaVA-1.5 and InstructBLIP backbones are demonstrated in Figure 10 and 11 respectively. In all plots, red fonts indicate the hallucinated contents, including any of object existence, attribute or relationship hallucinations. We can see that IKOD effectively mitigates object hallucinations without loss of details, which is consistent with the results in Table 2.

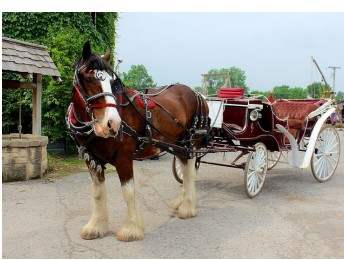

**IKOD:** The image features a brown horse pulling a red carriage with a white wheel. The horse is standing on a dirt road, and the carriage is parked next to a building. The horse is wearing a harness, and the carriage is attached to the horse's harness, indicating that it is being used for transportation. The horse and carriage are positioned in the foreground, and the scene is set in a dirt-covered area, creating a unique and picturesque setting.

**OPERA:** The image features a brown and white horse standing next to a horse-drawn carriage. The horse is positioned on the left side of the carriage, while the carriage is on the right side. The horse appears to be wearing a harness, indicating that it is ready to pull the carriage.\n\nIn addition to the horse and carriage, there are two people visible in the scene. One person is standing closer to the left side of the horse, while the other person is positioned near the right side of the carriage.

**VCD:** The image features a brown horse standing next to a red carriage, which is being used as a horse-drawn carriage. The horse is wearing a harness and is pulling the carriage, creating a picturesque scene. The carriage is large and spacious, providing ample room for the passengers.\n\nIn the background, there are a few trees, adding to the overall ambiance of the scene. The horse and carriage are the main focus of the image, showcasing the traditional mode of transportation.

**HALC:** The image features a brown horse with a white face, standing on a road next to a carriage. The horse is wearing a harness and appears to be pulling the carriage. The carriage is red and white, and it is positioned behind the horse.\n\nThere are two people in the scene, one standing closer to the horse and the other further away. The person near the horse seems to be attending to the horse or the carriage, while the other person is standing further back, possibly observing the scene or waiting for their turn to ride in the carriage.

**AGLA:** The image features a brown horse standing on a road, pulling a carriage with a red and white color scheme. The horse is positioned in the center of the scene, with the carriage behind it. The carriage has a bench seat, and the horse is wearing a harness, indicating that it is ready for a ride.\n\nIn addition to the horse and carriage, there are two people visible in the scene. One person is standing near the left side of the horse, while the other person is located on the right side of the horse. The presence of these individuals suggests that they might be passengers or caretakers of the horse and carriage.

- - - - - - - - - - - - - - - - - - - - - - - - - - - - - - - - - - - - - - - - - - - - -

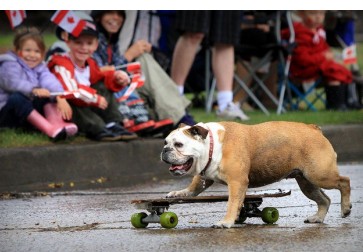

**IKOD:** The image features a brown and white dog skillfully riding a skateboard down a street, likely in a parade or a street event. The dog is the center of attention, as it appears to be the main attraction for the crowd of people watching the dog skate. There are several people in the scene, with some of them standing near the dog and others watching from a distance.\n\nThe crowd is spread out, with some standing close to the dog and others standing further back, observing the spectacle. The dog and the skateboard are positioned in the center of the scene, capturing the audience's attention and creating a lively atmosphere

**OPERA:** The image features a brown and white dog skillfully riding a skateboard down a street. The dog appears to be enjoying the ride and is the main focus of the scene.\n\nSeveral people are watching the dog from various positions along the street. Some are standing closer to the dog, while others are further away, observing the unique sight. A chair can be seen in the background, possibly belonging to one of the onlookers.\n\nIn addition to the people and the dog, there are a couple of handbags visible in the scene, likely belonging to some of the spectators.

**VCD:** The image features a brown and white dog skillfully riding a skateboard down a street. The dog appears to be enjoying the ride, and it is the main focus of the scene. \n\nThere are several people watching the dog from various positions along the street. Some are standing closer to the dog, while others are further away. A few chairs are scattered around the scene, with one near the left side of the image and another on the right side. \n\nIn addition to the people and chairs, there are a couple of handbags placed on the ground, one near the center of the image and the other closer to the right side.

**HALC:** The image features a brown and white dog skillfully riding a skateboard down a street. The dog appears to be enjoying the ride, and it is the main focus of the scene. \n\nThere are several people watching the dog's performance, with some standing closer to the dog and others further away. A few chairs are scattered around the area, possibly for the spectators to sit and enjoy the show. Additionally, there are a couple of umbrellas in the scene, one near the left side and another towards the right side, possibly providing shade for the onlookers.

**AGLA:** The image features a brown and white dog skillfully riding a skateboard down a street. The dog appears to be enjoying the ride, and it is the main focus of the scene. \n\nThere are several people watching the dog's performance from various positions around the street. Some are standing closer to the dog, while others are further away. A few chairs can be seen scattered around the area, possibly for spectators to sit and enjoy the show.\n\nIn addition to the people and chairs, there are a couple of handbags placed on the ground, possibly belonging to the onlookers. The overall atmosphere seems lively and entertaining as the dog showcases its talent on the skateboard.

Figure 10: Two examples of generated captions by different decoding methods with LLaVA-1.5 backbone.

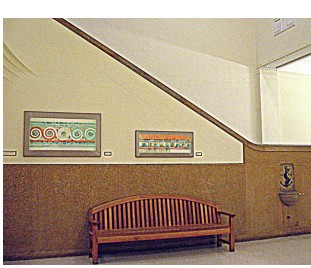

**IKOD:** The image features a long wooden bench placed in a room with a white wall and a stairway. The bench is placed in the center of the room, with a few steps leading up to the stairway. There are two paintings on the wall, one on the left and one on the right side of the stairway, adding to the room's aesthetic. The bench and the paintings are positioned in the room, creating a cozy and artistic atmosphere.

**OPERA:** The image depicts a spacious room with a wooden bench placed in front of a staircase. The bench is positioned near the bottom of the staircase, creating a cozy and inviting atmosphere. There are two paintings hanging on the wall above the bench, adding a touch of artistic flair to the space. In addition to the bench and paintings, there are several potted plants scattered throughout the room, giving it some greenery and life. A vase can be seen on the left side of the room, adding a decorative element to the space. Overall, the room exudes a warm and welcoming ambiance, perfect for relaxation or socializing.

**VCD:** The image features a long wooden bench placed in a room with a white wall and a staircase. The bench is situated in the middle of the room, with a potted plant placed on the left side of the bench. There are two paintings hanging on the wall, one on the left side and the other on the right side of the room. The paintings are positioned close to the bench, with one on the left and the other on the right side of the bench. The room has a spacious feel, with a lot of space between the bench and the paintings.

**HALC:** The image features a narrow staircase leading up to the second floor of an office building. Alongside the stairs, there is a wooden bench placed against the wall. A painting can be seen on one side of the room, while another painting hangs above the staircase landing. Two potted plants are also present in the scene, with one located near the bottom left corner and the other towards the top right corner. These elements create a cozy atmosphere within the space.

**AGLA:** The image features a long wooden bench placed in a room with a staircase. The bench is situated near the staircase, with a painting on the wall above it. There are two paintings on the wall, one on the left and one on the right side of the room. The bench is positioned in the center of the room, with a potted plant placed on the left side of the bench. The room has a clean and organized appearance, with a sense of elegance and sophistication.

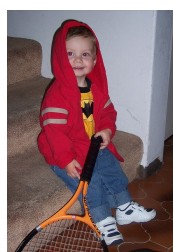

**IKOD** The image features a young boy sitting on a set of stairs, holding a tennis racket. He is wearing a red jacket and jeans, and appears to be in a happy and excited state. The tennis racket is positioned in his hand, ready to be used for a fun game or practice session. The boy is sitting on the stairs, and the tennis racket is placed in his hand, making it a great moment captured in the scene.

**OPERA:** The image features a young boy sitting on a set of stairs, holding a tennis racket. He is wearing a red hoodie and jeans, and appears to have a big smile on his face. The tennis racket is positioned close to the boy, indicating that he is ready to play or practice with it. There are several other objects in the scene, including a backpack and a handbag placed on the floor near the stairs. Additionally, a pair of shoes can be seen on the right side of the image.

**VCD:** The image features a young child sitting on a set of stairs, holding a tennis racket. The child is wearing a red jacket and jeans, and appears to be enjoying the moment. The tennis racket is positioned close to the child, and there is a shoe placed near the child's left foot. The scene is set in a house, with a chair placed nearby.

**HALC:** The image features a young boy sitting on the stairs, holding a tennis racket. He is wearing a red jacket and jeans, and appears to be enjoying his time playing with the tennis racket. A person can also be seen in the background, possibly accompanying the child or watching him play. There are several other objects scattered around the scene, including two shoes placed near the steps, one at the bottom of the staircase and another closer to the top. Another shoe can be found further up the stairs, close to the railing. Additionally, there is a handbag located towards the right side of the image, likely belonging to someone nearby.

**AGLA:** The image features a young boy sitting on a set of stairs, holding a tennis racket. He is wearing a red jacket and jeans, and appears to be enjoying the moment. The tennis racket is positioned close to the boy, and there are several other objects scattered around the scene, including a backpack and a shoe. The overall atmosphere is casual and relaxed, with the boy seemingly content with his tennis racket.

Figure 11: Two examples of generated captions by different decoding methods with InstructBLIP backbone.

