# OpenReview forum: "Fading Focus: Mitigating Visual Attention Degradation in Large Vision-Language Models"
_ICLR.cc/2025/Conference — ICLR 2025 Conference Withdrawn Submission_

### Official Review · Reviewer_e6HS · 2024-10-24

**Soundness:** 3
**Presentation:** 2
**Contribution:** 2
**Rating:** 3
**Confidence:** 4

**Summary:**

The paper addresses the issue of long-term bias in Large Vision-Language Models (LVLMs), where hallucination increases as the generated sequence length grows. To mitigate this, the authors propose IKOD, a method that uses KV merging to create shorter sequences focused on the image, which are then conditioned for next-token prediction. The logits from this process are combined with the original logits for final prediction. The method demonstrates modest improvements on vision-language benchmarks.

**Strengths:**

### S1. The paper is clearly written and easy to follow.

### S2. The proposed approach is straightforward and easy to understand, which enhances its accessibility.

**Weaknesses:**

### W1. The primary observation and motivation, which is that hallucination increases as sequence length grows (i.e., long-term bias), has been well-documented in prior works [1-5]. While the visualization in Figures 1-4 differs from previous studies, the core idea regarding long-term bias remains the same, and is not novel. Additionally, similar attention-based approaches have been explored in the past, as referenced in [2].

### W2. The improvements shown in Table 3 are minimal. The authors should have compared their method with more existing techniques to better establish the superiority of IKOD. Given the modest gains, multiple runs should have been conducted to validate the consistency of the improvements over the baseline.

### W3. The paper lacks a comparison with related work such as [3], which also involves attention modification to mitigate object hallucination in LVLMs. Although [3] does not directly target long-term bias, it raises an important point that higher attention does not always correlate with higher information content. However, in the method proposed here, tokens with lower image attention weights are selected as anchor tokens for aggregation. This potential conflict between approaches needs to be addressed. A discussion comparing these two methods is necessary to clarify their differences and prevent any confusion


### W4. Several figures, such as Figure 1 and Figure 4, are not self-explanatory. For instance, in Figure 1, the meaning of the x-axis and y-axis is unclear, as is the significance of the “top-20%” and “bottom-20%” tokens (the text refers to these as “first-20%” and “last-20%”). In Figure 4, the method for measuring hallucination is not well-explained.

### W5. While VILA and SVIT models are used in Figure 2, they are absent from the main experiments. It would strengthen the paper if InstructBLIP were included in Figure 2, and if the VILA and SVIT models were used as backbones in the experimental section for consistency.

### W6. It appears that the method is sensitive to the alpha value depending on the model used. Specifically, in the case of InstructBLIP, the impact of the proposed method seems minimal regardless of the alpha value, and performance may even degrade at higher alpha levels. An explanation for this behavior is needed. Additionally, a deeper analysis or statistics on which anchor tokens were selected during the KV merging process would be helpful to provide further insight into this issue.

[1] Analyzing and Mitigating Object Hallucination in Large Vision-Language Models, ICLR 2024.
[2] Mitigating hallucination in large multi-modal models via robust instruction tuning, ICLR 2024.
[3] Don't Miss the Forest for the Trees: Attentional Vision Calibration for Large Vision Language Models, Arxiv 2024.
[4] Multi-Modal Hallucination Control by Visual Information Grounding, CVPR 2024.
[5] VLind-Bench: Measuring Language Priors in Large Vision-Language Models, Arxiv 2024.

**Questions:**

### Q1. The experiments are limited to 7B models. Given the claim that the method is “highly scalable,” it would be useful to test the scalability of IKOD on larger models (e.g., 13B) to further validate this assertion.

---

### Official Review · Reviewer_BFhd · 2024-10-25

**Soundness:** 3
**Presentation:** 2
**Contribution:** 3
**Rating:** 6
**Confidence:** 4

**Summary:**

This paper proposes a decoding method for multi-modal large model. It shortens the length of key-value pairs (kv) during the decoding process by merging keys and values in the attention calculation. At the same time, it enhances the correlation between the kv sequence and visual features. This method shows a certain improvement in the evaluation sets for hallucinations.

**Strengths:**

The method  has a certain mitigating effect on hallucinations, with a noticeable improvement on POPE.

**Weaknesses:**

The inference process becomes relatively cumbersome.

**Questions:**

1. How much will the computational overhead of inference increase?

2. Will it have a significant impact on some problems that rely heavily on reasoning?

---

### Official Review · Reviewer_qmcc · 2024-10-26

**Soundness:** 3
**Presentation:** 2
**Contribution:** 2
**Rating:** 5
**Confidence:** 5

**Summary:**

The paper studies hallucination problems in large vision-language models. It proposes that hallucination can be caused by the model's diminishing attention to visual inputs as the number of generated tokens increases. The paper further introduces IKOD, a decoding strategy using key-value merging to address this problem. Experiments on POPE and COCO captioning datasets show that IKOD reduces hallucination.

**Strengths:**

1. This paper studies hallucination from the perspective of increasing sequence length, providing a potential view to explore the problem.
2. The proposed decoding strategy does not require training and shows promising results on hallucination benchmarks.

**Weaknesses:**

1. Figures are hard to understand.
  - Fig1: what axes represent layer or head? What does the color represent?
  - Fig2 is not referred to in the main text.
  - Fig5: the legends are confusing, does the orange and blue tokens also apply to KV cache?
2. How significant is the image attention diminishing? With longer sequences, the average attention score will also decrease. How can we identify the actual cause?
3. The conclusion that weak image attention leads to hallucination may not be valid. Fig.4 only shows that hallucinated tokens are prone to appear later in the sequence. For example, in Fig.4(a), there are also tokens with low image attention scores at the beginning of the sequence, but they do not cause much hallucination.
4. What are the lengths of image tokens and text tokens used in the experiments? In multi-modal models, there are usually more image tokens than text tokens. If that's the case, will an increase in generated text tokens significantly affect image attention?
5.  The design choice of KV merging needs further clarification. Why anchor tokens need to be chosen? If the motivation is to decrease the length, is it possible to use a uniform merging?

**Questions:**

1. Further experiments are needed to clarify that image attention really decreases with longer sequences. For example, how does text attention change w.r.t. sequence length? What is the average attention w.r.t. different sequence lengths? Compared with them, is image attention decrease similar or different?
2. The relationship between hallucination and image attention should be further clarified. Is hallucination really caused by weak image attention? Or is it actually relevant with sequence length? I'm also curious whether hallucination in LLMs has similar patterns with VLMs.
3. How does IKOD perform if it is applied to LLaVA1.5-HD? Will more image tokens affect the results?

---

### Official Review · Reviewer_9Djz · 2024-10-28

**Soundness:** 1
**Presentation:** 1
**Contribution:** 2
**Rating:** 3
**Confidence:** 4

**Summary:**

The paper introduces a novel decoding strategy for LVLMs, namely IKOD, a post-hoc and training-free strategy which relies on merging different tokens from a decoded sequence. The method is motivated by the observation that with an increase of decoding sequence length predicted tokens tend to hallucinate more as they do not rely on visual information. Authors evaluate their method on both: standard hallucination benchmarks as well as typical VQA benchmarks.

**Strengths:**

1. Presented method IKOD is novel and brings some gains on 2 standard hallucination benchmarks: POPE and CHAIR as well as VQA benchmarks.
2. Paper is generally well-motivated and the problem tackled clearly stated.

**Weaknesses:**

1. The main contribution which is the observation that LVLMs tend to hallucinate more with the length of a generated sequence is not novel and has been discussed in previous works e.g. M3ID [1] - which is also missing in this work.
2. The formal definitions are not very clear and in some parts incomplete. Therefore it makes it hard to understand IKOD in detail and appreciate the technical contribution of this paper.
3. In general the paper has multiple flaws in presentation and formulations. There are multiple typos and sentences without a meaning. It lacks clarity in several aspects as detailed in Questions section.
4. (minor) inconsistent references - multiple papers have their proceedings versions instead of arxiv. It is typically preferable to cite official proceedings.
E.g. CHAIR:
Anna Rohrbach, Lisa Anne Hendricks, Kaylee Burns, Trevor Darrell, and Kate Saenko. Object hallucination in image captioning. In Empirical Methods in Natural Language Processing (EMNLP), 2018

[1] Favero, Alessandro, et al. "Multi-modal hallucination control by visual information grounding." Proceedings of the IEEE/CVF Conference on Computer Vision and Pattern Recognition. 2024.

**Questions:**

- 2.1 (Eq 1) - xI not well-defined. By instruction I believe authors mean text features of instruction which is not obvious from this equation. Also it says ‘visual aligned features’ which is not true if we consider text instruction xI. L083 introduces a ‘language model’  which is then never used. Generally the formal presentation is incorrect and incomplete and I’d appreciate it to be more thorough.
- 2.2: is x from 2.1 different from the one here?
- L:125 should be an Eq with a (3)? Also Why xt in Qt formula and not Xt?
- Figure 1: I am not sure what exactly the plots represent? Are rows of different layers and columns of different tokens in a sequence length? By first 20% of tokens this excludes image tokens? By image attention, do authors mean attention to only image tokens? All of this is very unclear.
- There is never a reference in the text to Figure 2.
- It seems that Fig 2,3,4 are trying to present the same thing, is it correct?
- L217: ‘Besides, the hallucinated tokens are more concentrated in areas with low attention, which suggests lower image attention is more likely to cause the model to make errors.’ - how are the hallucinations denoted and detected here?
- L259: ‘Suppose the model has L layers in total, each with K heads.’ - which model do authors mean at this point? Image model or language model? Again, one could use more precise formulation here.
- L263: ‘S i t = P j Atti,j t [image_index], where image_index refers to the index of the image tokens’ - it would be beneficial to introduce a proper notation for ‘image tokens’ and therefore indices from the whole multimodal sequence.
- L268: It is unclear why the tokens with lower attention scores are expected to be more relevant to the query token.
- L292, 293: wasn’t K defined as the number of heads? What is the connection with top K tokens? Given this confusion I can’t decode Eq. (7).
- L304: is T now token sequence length? It used to be the number of layers just before? Or am I missing something?
- L340: why so?
- Table 1: IKOD brings gains, however I’d argue they are not significant. Could authors provide averages and standard deviations on multiple random seeds?
- Table 2: could authors provide details on how the Recall is obtained? Does it include synonyms defined in CHAIR benchmark? Does Avg. Len specify number of tokens or words or sth else?
- L421: Why did authors select the subset if COCO images and not the whole validation split as it’s been typically adopted in the literature?
- Table 4: could be good to provide references to other results (other columns).
- Table 5: seems there is a mistake and IKOD gives the worst results. If that's the case it’s important to discuss why.

---

### Official Review · Reviewer_5VeF · 2024-11-04

**Soundness:** 3
**Presentation:** 3
**Contribution:** 3
**Rating:** 6
**Confidence:** 4

**Summary:**

This paper introduces a training-free method called IKOD, designed to mitigate hallucinations in LVLMs. The approach generates logits from shorter sequences with enhanced image attention by merging key-value pairs. These logits are then combined with those obtained from the original decoding process, effectively reducing attention decay for visual tokens and minimizing hallucinations. Comprehensive experiments demonstrate the method's effectiveness in reducing hallucinations while maintaining strong performance on general benchmarks.

**Strengths:**

1. This method is training-free and does not require additional annotations, which is advantageous.

2. The underlying insight of the method is clear and straightforward, and it demonstrates strong performance.

**Weaknesses:**

This method requires an additional forward pass during inference, which will lead to extra inference computational cost.

**Questions:**

1. You state that a specific prompt is used for evaluating POPE and CHAIR. Does altering the prompt affect the results?

2. In Table 5, why does the low attention strategy perform the worst when the anchor ratio is set to 0.1?

3. You mention in Section 3.2 that InstructBLIP’s attention score is higher than that of LLaVA. However, as shown in Table 1, InstructBLIP has a lower F1 score than LLaVA 1.5, which suggests it experiences more hallucinations. Does this contradict your insight that a higher attention score is associated with fewer hallucinations?

---

### Note · Authors · 2024-11-13

I have read and agree with the venue's withdrawal policy on behalf of myself and my co-authors.